# Mother-Child Bond through Feeding: A Prospective Study including Neuroticism, Pregnancy Worries and Post-Traumatic Symptomatology

**DOI:** 10.3390/ijerph20032115

**Published:** 2023-01-24

**Authors:** Lorena Gutiérrez Hermoso, Patricia Catalá Mesón, Carmen Écija Gallardo, Dolores Marín Morales, Cecilia Peñacoba Puente

**Affiliations:** 1Department of Psychology, Universidad Rey Juan Carlos, Avenida de Atenas s/n, 28922 Alcorcón, Spain; 2Obstetric Department, Hospital Universitario de Fuenlabrada, Camino del Molino, 2, 28942 Fuenlabrada, Spain

**Keywords:** P-PTSD symptoms, mother-baby bonding, pregnancy, neuroticism, moderated-mediation model

## Abstract

Post-traumatic stress disorder (PTSD) is a common postpartum problem and influences maternal bonding with the infant. However, the relationship between this disorder, maternal personality, and the infant’s emotional state during feeding is not clear. The aim of the present study was to explore the contribution of neuroticism on the infant’s emotional state during feeding, by attending to the mediating role of postpartum PTSD (P-PTSD) symptoms and the moderating role of worries during pregnancy. A prospective design study was developed with 120 women with a low pregnancy risk. They responded to a questionnaire assessing maternal personality (first trimester), worries during pregnancy (third trimester), P-PTSD symptoms, and mother-baby bonding (4 months postpartum). The results showed a positive association among neuroticism, infant irritability during feeding, and P-PTSD symptoms, suggesting the latter plays a mediating role in the relationship between neuroticism and infant irritability (B = 0.102, standard error (SE) = 0.03, 95% coefficient interval (CI) [0.038, 0.176]). Excessive worries, related to coping with infant care, played a moderating role between neuroticism and P-PTSD symptoms (B = 0.413, SE = 0.084, *p* = 0.006, 95% CI [0.245, 0.581]). This relationship was interfered with by depressive symptoms in the first trimester (covariate) (B = 1.820, SE = 0.420, *p* = 0.016, ci [2.314, 0.251]). This study contributes to a better understanding of the role of neuroticism as an influential factor in the occurrence of P-PTSD symptoms, and in the impairment of infant bonding during feeding. Paying attention to these factors may favor the development of psychological support programs for mothers, with the aim of strengthening the bond with their child.

## 1. Introduction

Maternal bonding is the set of emotions and perceptions that the mother has towards her baby and towards herself in the new role of mother [1]. Despite the lack of conclusions from previous studies, it is believed that this bond originates during pregnancy and continues to develop immediately after birth and during the first months of the baby’s life [1,2]. Thus, the mother-infant bond is considered a dynamic emotional state that brings well-being to the newborn [2] and helps it to develop correctly at the cognitive and motor level [3,4].

It is argued that one of the factors that facilitates the maintenance and strengthening of maternal bonding is feeding [5]. Mothers often report that the experience of lactation is pleasurable and emotionally beneficial, since it helps them to physically contact their baby through play and feeding, as well as to connect emotionally with their infant through mutual eye contact [6].

However, this natural relationship that originates between the mother and her child, and that is maintained during feeding, can suffer alterations caused by feelings of dislike, resentment, or hatred towards the baby, by the need to give up the care of the baby, or by regret for being pregnant [7]. Previous research has shown that the etiology of bonding depends on the emotional stability of the mother during gestation and during the period of feeding the baby [8]. For example, studies comparing emotional behaviors of mothers toward their newborn infants during feeding, show that the differences in infant bonding are not due to the type of feeding (breastfeeding vs. bottle feeding) but to the emotional state of the mother [9,10]. 

It should be noted that childbirth is a complex experience that can sometimes be considered traumatic for the woman because of the threat it poses to her life and the life of her child [11]. Given these characteristics, post-traumatic stress disorder (PTSD) is prevalent in approximately 3% to 7% of women during the postpartum period (postpartum post-traumatic stress disorder, P-PTSD) [12,13]. In addition to exposure to a life-threatening event, PTSD is defined by the presence of different symptoms including intrusive memories, thoughts, or dreams that generate emotional distress and/or internal physiological reactions, and the need to strive to avoid memories, thoughts, emotions, situations, or people reminiscent of the traumatic event [14]. The P-PTSD prevalence is usually higher in those women who have suffered complications during pregnancy, during childbirth, after performing a cesarean section, or after having previously suffered a miscarriage [15,16]. In the case of non-risk pregnant women, previous literature points to pain experienced during childbirth, self-efficacy, locus of control, coping style, and fear of childbirth as predictors of the occurrence of P-PTSD symptoms [15,17]. 

In addition to clinical factors, there are psychological factors that play a role in maternal bonding and affect the likelihood of postpartum PTSD. To our knowledge, few studies have focused on the role of personality traits in the origin of mother-infant bonding during feeding, in terms of emotional distress after childbirth. Studies that have focused on the mother’s relationship with her baby, suggest that a personality defined by low persistence, avoidance tendency, low directionality, and presence of perfectionistic traits would be a risk motive for postpartum depression or deficits in bonding [18,19,20,21]. More specifically, neuroticism has been pointed out as the most influential personality factor in problems in the initiation and development of mother-child bonding. Neuroticism is a relatively stable personality trait that is defined by the tendency to experience negative emotions and to perceive any event as a great threat, so that people with a high level of neuroticism tend to have a greater predisposition to mood disorders [22]. Along these lines, there is a positive relationship between neuroticism and postpartum depression [23,24,25], and only one study has been able to confirm the relationship between neuroticism and PTSD-mediated attachment problems in pregnant diagnosed women [26]. However, there are no studies that confirm this relationship in women at low obstetric risk. 

Despite the results reported by Matatyahu Tahar et al. [26], to our knowledge, no studies have yet been conducted associating neuroticism with infant attachment disorder during feeding, P-PTSD symptom, and the role that maternal worries about pregnancy may play during the gestational period. Worry is a key variable in all anxiety disorders and is defined by an endless chain of defeatist thoughts about difficult situations to be faced in the future. In pregnant women, these worries usually originate during gestation, and their content is related to childbirth or possible complications in the development of the fetus [27], so they usually maintain a positive relationship with neuroticism [27,28] and postpartum depression [29] according to previous research. Given the paucity of studies on postpartum PTSD symptoms in women at low obstetric risk, this study aimed to integrate neuroticism, as a risk personality variable, into a model that considers P-PTSD symptoms, and worries during pregnancy, as factors that may be related to infant attachment problems during feeding. Specifically, we propose that greater neuroticism is associated with greater infant bonding problems in feeding through the mediating effect of P-PTSD symptoms (Hypothesis 1). On the other hand, within the previous model proposed (neuroticism-P-PTSD symptoms-infant bonding in feeding), we hypothesize that worries during pregnancy could exert a moderating effect between neuroticism and P-PTSD symptoms (Hypothesis 2).

## 2. Materials and Methods

### 2.1. Participants and Procedures

The data used in the present study come from prospective research on health in pregnant women conducted in a hospital context (ommited for blinded review). In the present study, an analysis was conducted on the assessment of infant bonding during feeding, and the presence of PTSD symptoms 4 months postpartum based on the assessment of the mother’s personality traits in the first trimester of pregnancy. Additionally, measures on worries during the third trimester of pregnancy were also performed. Therefore, assessments were conducted at three time points: first and third trimester of pregnancy, and 4 months postpartum. Inclusion criteria were being over 18 years of age, being autonomous to adequately answer the questions on the forms in Spanish, not being diagnosed with any psychiatric disorder during pregnancy, not presenting medical complications in the fetus that could put the pregnancy at risk, and completing all the questionnaires administered at each time point. Exclusion criteria were miscarriage and neonatal alterations diagnosed at birth.

As shown in Table 1, a total of 290 pregnant women completed questionnaires during the first trimester at the obstetric visit with the midwife. However, 5 questionnaires were excluded because 5 women suffered a miscarriage, therefore, the final sample in the first trimester was 285 participants. In the third trimester, the sample was 123 participants and, finally, the final postpartum sample consisted of 120 women who completed all the questionnaires. Both the third trimester and postpartum questionnaires were sent by mail.

The mean age of the participants (*n* = 120) was 31.28 years (SD = 3.96), ranging from 23 to 42 years. Most of them (85%, *n* = 102) had planned their pregnancy and 51.6% (*n* = 62) were first-time mothers. Regarding employment status, 67.5% (*n* = 81) were working. Regarding educational level, according to The International Standard Classification of Education (ISCED, 2011), half of the participants (*n*= 58, 48.3%) had Bachelor’s degrees or equivalent studies (level 6 ISCED), 34 women (28.3%) had Master’s degrees or equivalent studies (level 7 ISCED), and 28 women (23.3%) had lower secondary education (level 2 ISCED). 

Regarding the type of delivery, 73 women had a vaginal delivery (57.7%), 28 women had a cesarean section (25.2%), and 19 women had an instrumental delivery by forceps or vacuum extraction (17.1%). Epidural was the type of anesthesia chosen by 88 women (73.3%), 14 women chose spinal anesthesia (11.7%), 8 women chose local anesthesia (6.7%), and 10 women did not choose any type of anesthesia (8.3%). As for the Apgar score (degree of infant tolerance at birth), the mean score was 8.70 in the first minute of life (SD = 1.04) and 9.80 at five minutes of newborn life (SD = 0.54).

Finally, on the postpartum period, 62 women continued breastfeeding four months after delivery (51.7%) and 58 women fed with bottle-feeding (48.3%). No statistically significant differences were observed in any of the variables (sociodemographic and obstetric) mentioned above between the sample that only participated in the first trimester (*n* = 285) and the one that finally concluded the study (*n* = 120). 

### 2.2. Ethics

The present study is part of a larger project that received approval from the Human Research Ethics Committee (reference omitted for blinded review). Prior to the evaluation, the participants signed the informed consent form and the study participation form. In these documents, all the procedures to be carried out in the study were detailed. In addition, the researchers responsible for the project informed the participants that there were no negative consequences for their health or for the health of the baby and that they could withdraw their consent whenever they considered it appropriate.

### 2.3. Measures

#### 2.3.1. Sociodemographic and Clinical Data

Sociodemographic data such as the participants’ age, educational level, and employment status were collected by an ad-hoc questionnaire. Clinical data (i.e., previous deliveries, planned pregnancy, previous miscarriages, apgar test, type of delivery and anesthesia during delivery) were obtained from the obstetric history through the electronic medical records provided by the Gynecology and Obstetrics unit where the pregnant woman was cared for. 

#### 2.3.2. First Trimester of Pregnancy

##### Neuroticism

For this study, we used the Spanish reduced version of NEO Personality Inventory [30]. This questionnaire evaluates the 5 personality factors: neuroticism, extraversion, agreeableness, responsibility, and openness to experience. The responses to this questionnaire are distributed on a Likert-type scale comprising scores ranging from 0 (“completely disagree”) to 4 (“completely agree”). Each factor is composed of 12 items, with a theoretical range from 0 to 48 according to its factor structure [31]. For this study, we employed only the neuroticism dimension (Cronbach’s alpha (α) = 0.80).

#### 2.3.3. Third Trimester of Pregnancy

##### Pregnancy Worries

Pregnancy worries at the third trimester of pregnancy were assessed by the Spanish version of the Cambridge Pregnancy Worry Scale (CWS) [32]. This instrument allows the assessment of 17 situations that pose a specific concern during pregnancy. Women must rate their level of concern by means of a 6-point Likert scale, ranging from 0 (“not a concern”) to 5 (“a major concern”). Its factorial structure reveals four factors: worries related to the pregnant woman’s own health, those related to socio-health aspects, those related to socio-economic aspects, and those related to family and couple relationships. In our data, CWS shows good reliability, with α ranging from 0.70 to 0.79 for the different factors and α = 0.81 for CWS-total.

For the present study, we performed a first correlational analysis of the four general factors with the rest of the variables proposed in our model. In a second step, a correlational analysis was carried out between the specific worries contained in the factors that had been found to be significant in the previous step and the rest of the variables of the proposed model (neuroticism, P-PTSD symptoms and maternal bonding during feeding).

#### 2.3.4. Four Months Postpartum

##### P-PTSD Symptoms

PTSD symptoms in the postpartum were assessed using the Spanish version of the Modified Perinatal PTSD Questionnaire [33]. The scale consists of 14 items that assess the re-experiencing of symptoms such as intrusions, arousal, or excessive avoidance of circumstances related to childbirth or pregnancy, with higher scores indicating higher levels of post-traumatic stress. There was a good internal consistency in the participant sample (α = 0.84).

##### Mother-Baby Bonding during Feeding

Mothers’ perceptions of their bond with their baby were assessed by means of the self-report Mother and Baby Scales (MABS) [34]. The MABS assess the mother’s perceptions of her relationship with her newborn. These perceptions can be grouped into two factors: perceptions related to the infant’s behavior and perceptions related to the mother’s confidence in caring for her infant. For this study, the first factor was selected, which is composed of 39 items divided into 5 dimensions: alert-interest (8 items *“when I talk to my baby he seems to pay attention”*), unstable-irregular (15 items *“my baby has become restless”*), irritable during feeding (8 items *“during feedings my baby has tended to become restless or cry”*), alert during feeding (5 items, *“after feedings, my baby’s mood has been awake and alert”*) and easy temperament (3 items, *“in general, how difficult is your baby?”*). Items are presented in a 6-point Likert-type response format ranging from 0 (“not at all/never”) to 5 (“a lot/frequently”) for the dimensions alert-interest, unstable-irregular, irritable during feeding, and alert during feeding, respectively, and 7 points for easy temperament. According to our aim (i.e., exploring perceptions of bonding during feeding), infant irritability dimension and alertness during feeding dimension were selected. Both scales showed adequate Cronbach’s coefficients (α = 0.72 and α = 0.75, respectively).

#### 2.3.5. Covariates

According to previous literature, P-PTSD symptoms may be influenced by the type of delivery [15,16], so in the present study, the type of delivery, the type of anesthesia chosen at delivery, and the Apgar test score in the first and fifth minute of life of the newborn were included as covariates. Age and feeding choice were also included as covariates to control for their possible effect on the model. Likewise, given that previous studies point to a high rate of comorbidity of PTSD symptoms with depressive and anxious symptomatology [35,36], we also considered both symptoms (at first trimester) as covariates. 

To evaluate anxious and depressive symptomatology during first trimester, the Spanish validation of the Symptom Checklist-90-4 (SCL-90-R) [37] was chosen. This instrument consists of 90 items that evaluate 9 dimensions on a 5-point Likert-type scale: somatization (12 items), obsessive compulsiveness (10 items), interpersonal sensitivity (9 items), depression (13 items), anxiety (10 items), hostility (6 items), phobic anxiety (7 items), paranoid ideation (6 items), and psychoticism (10 items). SCL-90-R shows adequate psychometric properties [38]. For the present study, depression (13 items) and anxiety (10 items) scales were employed. In our data, Cronbach’s alpha values are 0.85 for depression and 0.82 for anxiety.

### 2.4. Statistical Analysis

Data was analyzed using SPSS version 27.0 statistical software. First, descriptive analyses were used to obtain the frequencies, percentages, means, and standard deviations of the clinical and sociodemographic characteristics of the participants. In parallel, reliability and internal consistency analyses were performed by means of Cronbach’s alpha coefficient (α). Finally, correlation analyses were performed to explore the existing relationships between the variables of interest by means of Pearson’s correlation. Those relationships between variables with *p* < 0.05 were considered significant. In relation to this type of analysis, a first step aimed to explore the significant relationships between the variables under study in terms of the four general factors on worries during pregnancy. Based on the significant relationships found, the second step was established to explore the correlations among the same variables according to the specific worries of the general factor that was found to be significant.

Once the significant correlations between the variables were confirmed, our first hypothesis is supported by the mediating effect of P-PTSD symptoms. For this purpose, model 4, provided by the PROCESS macro version 3.5. of SPSS software, was tested [39]. Finally, the moderated-mediation model was tested using model 7. Neuroticism is proposed as an independent variable predicting maternal perceptions of infant behavior during feeding. This relationship is mediated by P-PTSD symptoms and influenced by the moderating effect of worries during the third trimester of pregnancy (Figure 1). The aforementioned covariates were included in the models. Statistical significance was defined as a two-tailed *p*-value of <0.01. To test its statistical significance, the bootstrapping method, with 5000 bootstrap samples, was used to construct the 95% confidence intervals.

## 3. Results

### 3.1. Correlations among Variables (Considering All Four Worry Factors)

Neuroticism was positively and significantly correlated with P-PTSD symptoms (*r* = 0.428, *p* < 0.001) and with infant irritability (*r* = 0.217, *p* = 0.020). With respect to the relationship of neuroticism to worries during pregnancy, it correlated significantly with worries about the environment (*r* = 0.305, *p* = 0.001), with worries related to socio-economic aspects (*r* = 0.204, *p* = 0.024), and to socio-health aspects (*r* = 0.311, *p <* 0.001) and health-related worries (*r* = 0.391, *p <* 0.001). 

Regarding P-PTSD symptoms, they were significantly related to socioeconomic worries (*r* = 0.281, *p* = 0.008) and to socio-health worries (*r* = 0.247, *p* = 0.021); in relation to bonding, P-PTSD symptoms maintained a significant relationship with infant irritability (*r* = *0*.362, *p <* 0.001) and with the baby’s alertness during feedings (*r* = −0.311, *p <* 0.001). Finally, only infant irritability in feeding correlated positively and significantly with socio-health worries (*r* = 0.227, *p* = 0.036).

### 3.2. Correlations among Variables (Considering Specific Worries Contained in the Statistically Significant Worry Factors)

From the previous results, we concluded that the socio-health worries factor is the only one that maintains significant relationships both with neuroticism (independent variable) and with the variables related to the baby’s bonding with the mother during feeding (dependent variable) and with P-PTSD symptoms (mediator). Thus, we set out to specifically analyze the correlations between each of the worries that encompass this factor and the rest of the variables of the model. That is, worries related to childbirth, hospital visits, medical examinations, and coping with infant care. The results (Table 2) show that only worries related to infant care coping maintain significant relationships with neuroticism (*p* < 0.001), P-PTSD symptoms (*p* = 0.012) and infant irritability during feeding (*p* < 0.001). Therefore, according to Hayes [39], these worries are proposed as moderators of the relationship between neuroticism on bonding during feeding (infant irritability) attending to the presence of P-PTSD symptoms as a mediator.

### 3.3. Testing the Mediation Effect of P-PTSD Symptoms between Neuroticism and Infant Irritability

We analyzed the mediating role of P-PTSD symptoms in the relationship between neuroticism and infant irritability during feeding. As Table 3 shows, the results first revealed that neuroticism was positively associated with infant irritability (total effect model of X→Y, *c* path) with a significant unstandardized regression coefficient (B = 0.165, standard error (SE) = 0.06, *p* = 0.020, 95% coefficient interval (CI) [0.021, 0.280]). Second, the findings suggested that neuroticism was positively correlated with P-PTSD symptoms (*a* path; B = 0.431, SE = 0.085, *p* = 0.000, 95% CI [0.262, 0.600]) and third, a significant positive association was confirmed between P-PTSD symptoms and infant irritability during feeding (*b* path; B = 0.273, SE = 0.072, *p* = 0.011, 95% CI [0.975, 0.377]). In addition, the findings implied that the residual direct association between neuroticism and infant irritability during feeding was significant (*c’* path; B = 0.053, SE = 0.070, *p* = 0.044, 95% CI [0.087, 0.192]). On the other hand, the bootstrapping analysis confirmed hypothesis 1: neuroticism correlates to infant irritability during feeding partially through P-PTSD symptoms (B = 0.431, SE = 0.02, 95% CI [0.296, 0.574]). The indirect effect accounts for 16% of the total influence of neuroticism on infant irritability during feeding.

### 3.4. Testing the Moderated-Mediation Model Neuroticism- P-PTSD Symptom-Infant Irritability (Worries as Moderator)

Next, we examined the potential conditional effects of worries related to coping in caring for the baby (third trimester of pregnancy) on this mediation analysis. As can be seen in Table 4, hypothesis 2, regarding the moderating effect of worries, was supported based on the significant association between the interaction term (Neuroticism x worries related to coping in caring for the baby) and P-PTSD symptoms (B = 0.136, SE = 0.045, *p* = 0.046, 95% CI [0.077, 0.158]). The results show that neuroticism had a significant impact on infant irritability during feeding through the effect of P-PTSD symptoms in women who report high worries (during the third trimester) about coping with infant care (B = 0.413, SE = 0.084, *p* = 0.006, 95% CI [0.245, 0.581]). This relationship is not statistically significant in women who report medium worries about coping with the baby (B = 0.301, SE = 0.120, *p* = 0.065, 95% CI [0.333, 0.801]), nor in women who report low worries (B = 0.257, SE = 0.110, *p* = 0.061, 95% CI [0.038, 0.476]) (see Table 4). Regarding the effect of the aforementioned covariates, a significant effect of depressive symptoms during the first trimester was observed (B = 1.820, SE = 0.420, *p* = 0.016, 95% CI [2.314, 0.251]).

The proposed model explains 20% of the variance in infant irritability during feeding (*R^2^* = 0.201, *p* = 0.006) (see Table 4). 

## 4. Discussion

The quality of the maternal relationship during feeding has important implications for maternal caregiving experiences and infant developmental outcomes [40,41]. For this reason, using data provided by assessments of pregnant women during gestation and postpartum, and relying on theoretical models of maternal attachment, this study aimed to explore those psychological factors that may impact the development of the bond between mother and infant. The results supported our initial hypotheses of the effect of neuroticism on bonding through the feed, and that this association would be mediated by postpartum PTSD symptoms and moderated by worries during pregnancy. It is important to note that, even though the sample could have been larger, the prospective design of the study, and the direct observation of the data collected, provides significant value in understanding the effects of the mother’s personal characteristics during gestation on maternal perception of the infant’s mood during feeding. Given the complexity of sample access in low obstetric risk pregnant women, the findings found in the present study merit consideration to adequately address those psychosocial factors that influence early mother-infant interactions during feeding.

The relation of postpartum PTSD with anxiety and depressive symptoms during gestation is well established. A previous study observed that elevated levels of anxiety and depression during pregnancy predicted postpartum PTSD symptoms due to perceived lack of control over childbearing [42]. Another study confirmed that previous psychopathology during the first trimester was also a predictor of the onset of PTSD symptoms [43]. However, only two studies reported significant results on the relationship between personality traits and postpartum PTSD symptoms, confirming that mothers with high levels of neuroticism were at greater risk for PTSD both in pregnancy and after delivery [26,44]. The results of our study add to the limited existing literature, reporting a predictor effect of neuroticism on PTSD symptom reported four months after delivery. Furthermore, attending to the limitations offered by previous literature, we controlled for the effect of emotional state given that it may influence personal characteristics [24,45]. Along these lines, we detected that it is the depressive symptoms identified during the first trimester of pregnancy that influence the relationship between neuroticism and the probability of the occurrence of postpartum PTSD symptoms. These data may represent an interesting finding that will enable health care professionals to provide appropriate strategies to adequately regulate emotional states associated with childbearing, based on screening for personality traits (i.e., neuroticism) that are associated with certain postpartum psychopathologies. 

Compared to other prevalent postpartum emotional problems, such as depression, the literature regarding postpartum PTSD and bonding is limited, with a small number of studies reporting negative effects of PTSD symptomatology and mediated by emotional symptoms [46,47]. However, to our knowledge, there are no studies exploring the role of cognitive factors in the effect of PTSD symptomatology on maternal-filial difficulties. Along these lines, this is the first study to explore worries, as one of the main characteristics of people with a high level of neuroticism, in the effect of this personality trait on PTSD symptoms and bonding problems. Worries are universal in all pregnant women. In a previous study, it was reported that the content of worries varies according to the trimester of pregnancy. Thus, those concerns related to the health of the fetus are the most frequent in the first trimester, while their intensity is lower throughout the third trimester [48]. Supporting these results, our study emphasizes that those concerns related to the mother’s visualization of herself exercising caring behaviors towards her baby are the most relevant, having an effect on PTSD symptoms after childbirth.

A tendency to ruminate is a major characteristic of people with high neuroticism scores. In addition, cognitive rumination is critical for a wide range of undesirable outcomes for both the baby and the mother (prematurity, low birth weight, lower Apgar scores, vaginal bleeding, or risk of miscarriage) [49]. Based on evidence from clinical practice, the cognitive interpretation of pregnancy as a potentially threatening event can generate a multitude of negative health consequences. Therefore, one of the focuses of therapeutic interventions with women at low obstetric risk should be on the management of maladjusted interpretations related to the care of the baby.

While neuroticism was significantly correlated with, and predicted, an infant’s irritability state during the feed, this association was entirely mediated by the effect of postpartum PTSD symptoms. Presumably, these early results suggest that mothers who feel insecure, and tend toward instability in managing their emotions, may have difficulties in bonding and may transmit emotional instability to their infant due to the traumatic event that childbirth may entail for them. The indirect pathway of the model through the mediation of postpartum PTSD highlights the need to improve the woman’s understanding of the processing and integration of her pregnancy experience to mitigate the negative effects on maternal bonding with the baby and parenting style by addressing certain personal traits considered as risk factors.

A fact of interest in relation to our correlational results is that neither neuroticism nor PTSD symptoms after childbirth were significantly related to the positive dimension of maternal bonding (alertness during feeding). Therefore, we could not confirm that therapeutic interventions on the emotional instability of the mother during pregnancy could have a beneficial effect on the positive shared experiences of the baby and the mother during feeding. This data is related to the premises on which Cognitive Psychology and Positive Psychology are based. The former focuses on the reduction of negative cognitive-emotional variables (misaligned emotions or interpretations) as the main indicator of improvement in people’s quality of life [50]; on the contrary, Positive Psychology is based on the improvement of positive psychological aspects of people, instead of attending to the reduction of negative characteristics (i.e., neuroticism), with the intention of exerting a positive effect on perceived well-being or quality of life [51]. Based on the nature of the variables in our model, Cognitive Psychology can help pregnant women to break the vicious cycle “thought-emotion-behavior” by illustrating how inappropriate thoughts can have unfavorable results on the bond established. By applying techniques based on this stream, women can identify their distortions regarding a lack of capacity about baby care, and replace them with more efficient and positive ways of thinking. Such a change gives individuals the opportunity to form a new, more appropriate interpretation of themselves and the world, that leads to the activation of personal resources geared to the demands of parenting.

### 4.1. Limitations

First, it is worth noting the difficulty of generalizing the results, as it is a convenience sample from a single hospital. Likewise, since it is a sample of low obstetric risk, with some similarity regarding its characteristics, the limitations of the interpretation and extrapolation of the results must be taken into account. Sample loss in the different phases of the study could also bias the interpretation of the results, although the sample loss is similar to that found in prospective studies of this nature, and no statistically significant differences were found in the sociodemographic and obstetric variables between the pregnant women who completed this study and those who did not. The self-reported nature of the instruments used in the assessment may bias the veracity of the information offered by the participants on the outcome variables. Regarding the multivariate analysis, the effect of clinical factors such as type of delivery or choice-feeding was considered within the model. Nevertheless, although in the present research the control of demographic and clinical variables associated with PTSD symptoms in pregnancy has been considered, the effect of other physical symptoms related to PTSD should also be controlled. For example, physical symptoms such as pain during pregnancy or childbirth could be related to traumatic experiences according to the findings established by some previous studies [52]. We have confirmed the importance of the management of caregiving worries as a significant moderator between emotional symptomatology in the prenatal stage and its implication in the bonding in the postnatal stage. However, within a broader family context, it would be interesting to consider other variables, such as the role of the father figure in the bond. Several previous studies have found that the mother’s neuroticism was related to a lower parental attachment and greater tendency to control stressful situations with respect to infant care [53,54]. On the other hand, socioeconomic variables, such as maternity leave or financial resources, have been related to the sense of competence in the development of tasks typical of the mother’s role and, consequently, with the possibility of developing an adequate bond with the baby. The leave time, financial resources, as well as the help in caring for the baby by the partner or other relatives should be considered in future studies to gain in-depth knowledge of all the factors that can influence the mother-baby bond [55,56]. Therefore, although our results should be interpreted with caution, we propose that this model can be taken as a basis for future studies that propose possible explanatory mediators between maternal personality, postpartum PTSD, and the difficulties manifested by the infant in the relationship with his or her mother. 

### 4.2. Implications for Clinical Practice, Policy and Research

The current findings have implications for improving the emotional health of mothers with low obstetric risk. For example, the inclusion of psychological care from the first ultrasound could mitigate rumination about the ability to care for the baby. As has been shown, the mechanism of rumination is linked to profiles with a high level of neuroticism, which increases the probability of suffering emotional problems during pregnancy, in the perinatal stage, or affecting the relationship that the mother establishes with her child. Therapies based on cognitive-behavioral techniques, or acceptance and commitment to motherhood, are considered the most effective in the treatment of ambivalence, mood swings, or depressive symptoms in the face of new personal, psychological, and social changes in women [57,58]. Facilitating appropriate strategies focused on engaged action to adequately regulate these changes could significantly limit the occurrence of postpartum PTSD and thus strengthen maternal-filial relationships. As a future line, we propose the inclusion of the study of oxytocin levels during pregnancy as one of the most influential biological factors in promoting maternal bonding [59], along with the personality and emotional factors included in our model. A recently published study has confirmed that stable levels of oxytocin after delivery are related to resilience to maternal stress, the expression of positive affection, and the development of thoughts linked to secure attachment with the infant [60]. Therefore, it is important to note that hormonal variations, specifically oxytocin during pregnancy and postpartum, may play a role in the emergence of typical bonding behaviors and mental representations in the mother.

## 5. Conclusions

In this study, we found that a high level of neuroticism in low obstetric risk pregnant women, can seriously affect the bond established with the infant. In addition, pregnant women with more pronounced caregiving and parenting worries showed significant indirect effects of neuroticism on their perceptions of their children’s difficulties in established maternal relationships four months after childbirth. This relationship is explained by postpartum PTSD symptoms, showing that unresolved ruminations during pregnancy accentuate traumatic perceptions of pregnancy or childbirth when women have a personality profile characterized by a tendency to ruminate and emotional instability. These results suggest clinical implications for identifying the most vulnerable mothers, with the intention of adapting psychological support programs aimed at preventing mental health problems during pregnancy and in the perinatal stage.

## Figures and Tables

**Figure 1 ijerph-20-02115-f001:**
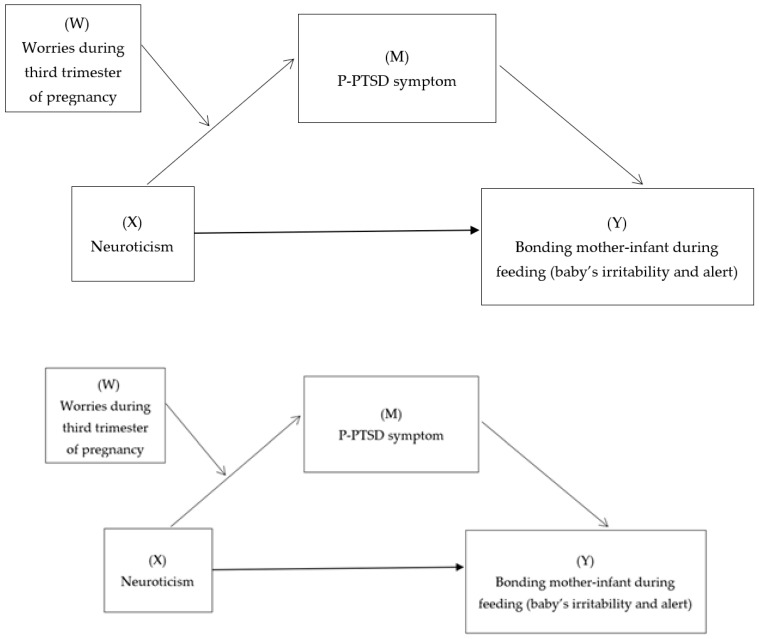
Conceptual moderated-mediation model.

**Table 1 ijerph-20-02115-t001:** Research design process.

Time	Participants	Participants Excluded	Percentage of Excluded	Reasons
Assessed for eligibility	290	5	1.72%	Miscarriage (*n* = 5)
First trimester	285	162	56.84%	Not complete questionnaire (*n* = 157) Miscarriage (*n* = 5)
Third trimester	123	3	2.43%	Not complete questionnaire (*n* = 3)
Postpartum	120	0	0%	
Analysis	120	0	0%	

**Table 2 ijerph-20-02115-t002:** Descriptive statistics and bivariate correlations.

	M (SD)	1	2	3	4	5	6	7	8
1.Neuroticism	18.05 (7.26)	--							
2.Worries about giving birth	2.22 (1.26)	0.129	--						
3.Worries related to hospital visits	0.74 (0.06)	0.143	0.023	--					
4.Worries related to medical examinations	1.05 (1.00)	0.274 **	0.229 *	0.553 **	--				
5.Worries related to coping in caring for the baby	1.85 (0.51)	0.316 *	0.469 **	0.239 **	0.369 **	--			
6. P-PTSD symptoms	7.90 (0.50)	0.428 **	0.116	0.010	0.139	0.272 *	--		
7.Irritability during feeding	7.02 (2.38)	0.217 *	0.179	−0.034	−0.095	0.388 **	0.362 **	--	
8.Alertness during feeding	27.84 (11.02)	−0.128	−0.035	−0.009	−0.072	−0.161	−0.311 **	0.540 **	--

Note: * *p* < 0.05; ** *p* < 0.001.

**Table 3 ijerph-20-02115-t003:** Mediation effect of P-PTSD symptoms on infant irritability during feeding.

Variables	*B*	*SE*	*t*	*p*	*LLCI*	*ULCI*
Direct and total effect *a* path, Neuroticism → P-PTSD symptoms	0.431	0.085	5.075	<0.001	0.262	0.600
*b* path, P-PTSD symptoms → Infant irritability during feeding	0.273	0.072	3.364	0.011	0.975	0.377
*c’* path, Neuroticism → Infant irritability during feeding (Direct effect)	0.053	0.070	1.76	0.044	0.087	0.192
*c* path, Neuroticism → Infant irritability during feeding (Total effect)	0.165	0.06	2.35	0.020	0.021	0.280
Indirect effect of Neuroticism (X) on Infant irritability during feeding (Y) through P-PTSD symptoms	0.102	0.035			0.038	0.176

**Table 4 ijerph-20-02115-t004:** Moderated-mediation model of neuroticism on infant irritability during feeding through P-PTSD symptoms (mediator), taking into account worries related to coping in caring for the baby (moderator).

Predictor	Infant Irritability during Feeding					
	*B*	*SE*	*t*	*p*	*LLCI*	*ULCI*
Neuroticism	0.059	0.153	0.873	0.045	0.256	0.355
Neuroticism x Worries related to coping in caring for the baby	0.136	0.045	2.029	0.046	0.077	0.158
Postpartum PTSD symptoms	0.207	0.199	1.044	0.003	0.197	0.377
[Model R = 0.391						
R^2^ = 0.201 F = 8.858 *p* = 0.006]						
Effect of covariates						
Type of delivery	0.665	1.034	1.221	0.240	−3.451	0.921
Choice-feeding	−0.612	1.047	0.813	0.430	−11.820	5.242
Type of anesthesia	−0.751	1.723	0.570	0.580	−2.663	4.625
First minute Apgar test	0.659	0.340	0.289	0.776	−2.438	3.213
Fifth minute Apgar test	0.296	0.951	0.100	0.922	−5.923	6.520
Depressive symptoms first trimester	1.820	0.420	2.49	0.016	2.314	0.251
Anxious symptoms first trimester	1.262	1.515	0.058	0.563	−5.577	3.058
Age	−0.124	0.184	−0.675	0.502	−0.491	0.242
Conditional indirect effect of Neuroticism (X) on Infant irritability during feeding (Y) by different levels of worries related to coping in caring for the baby (W) through the effect of Postpartum PTSD symptoms (M)						
Low level of worries	0.257	0.110	2.332	0.061	0.038	0.476
Medium level of worries	0.301	0.120	3.874	0.065	0.333	0.801
High level of worries	0.413	0.084	4.783	0.006	0.245	0.581
Indirect effect of Neuroticism (X) on Infant irritability during feeding (Y) through Postpartum PTSD (M)	0.061	0.033			0.004	0.130

Note. *LLCI* = lower limit 95% confidence interval; *ULCI* = upper limit 95% confidence interval (bias-corrected bootstrap confidence intervals).

## Data Availability

The data presented in this study are available on request from the corresponding author.

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
