# Peer review of "Mother-Child Bond through Feeding: A Prospective Study including Neuroticism, Pregnancy Worries and Post-Traumatic Symptomatology"

_ijerph, 2023, doi:10.3390/ijerph20032115_

Round 1

Reviewer 1 Report

Review

This prospective study describes 120 women in different stages of pregnancy and postpartum (first and third trimester, after delivery and 4 months postpartum) to assess the relationship between maternal neuroticism, infant’s emotional state during feeding, PTSD and worries during pregnancy.

Thank you for the opportunity to revise this interesting manuscript. There are several fundamental issues in the manuscript that I think you should address.

First – throughout the manuscript you sometimes refer to feeding and sometimes refer to breastfeeding. I assume this is a mistake and you mean feeding (due to half your sample that is not breastfeeding). However, this is very confusing to me as a reader and should be consistent throughout the manuscript.

A similar problem is “posttraumatic symptomatology”. You should explain what this term means. Sometimes you refer to it as PTSD and sometimes as posttraumatic symptomatology. This again is confusing to the reader. Please stay with one terminology throughout the manuscript.

Second – and this is the most important comment: you state in the introduction 2 aims for your study. The first, the association between neuroticism and infant bonding problems in feeding. The second is that worries during pregnancy have a moderating effect on the association between neuroticism and the presence of PTSD (posttraumatic symptomatology) after delivery. However, in your manuscript, you mainly focus on the second aim. Even in the conclusion, there is mainly a focus on the second aim of the study. This problem should be addressed throughout the manuscript. In order to help readers understand the flow of your manuscript subheadings may help and I think the figures should be better planned to facilitate readers understanding the models you suggest (results 3.2, 3.3 and 3.4 should be explained in better figures).

INTRODUCTION

In lines 48-49, this sentence that associates bonding with feeding should be elaborated a little bit. The second sentence in this paragraph, line 51, why did you choose ‘therefore’?

In line 65, please explain what is posttraumatic symptomatology.

MATERIAL AND METHODS

285 women participated in the first part of the study (and completed the questionnaire in the first trimester). Only 120 remained in the final sample. Please give a comparison between the original cohort and the final sample. This is a possible source for major selection bias in your study.

In lines 126-127 – what did you mean secondary and primary studies? This is not clear.

Figure 1 – 3 women were excluded (did not complete the …[word missing])

Line 177 – obstetric history – how was this obtained? Through the electronic medical records?

Figure 3 – I think it is not clear and does not add a lot to the manuscript.

DISCUSSION

Line 408 – groundbreaking – I suggest using a more subtle word.

Line 467 – I suggest deleting the first sentence (‘our study is not without limitations’). I suggest adding the selection bias (120 women out of 285 in the original cohort).

CONCLUSION

Where is a statement regarding the bonding problem during feeding (aim 1)?

Author Response

Dear Evelin Li,

We would like to thank you for your interest in our manuscript entitled Mother-child bond through feeding. A prospective study including neuroticism, pregnancy worries and post-traumatic symptomatology (ijerph-2098508) in the special issue of Maternal Perinatal Mental Health 2nd Edition of the Women's Health section. We appreciate the time that you and the other reviewers have dedicated to reading the manuscript and providing suggestions. Your suggestions have enriched the manuscript considerably. Likewise, we have incorporated all the comments suggested. Following your directions, we have proceeded to revise our manuscript, highlighting the changes by using the track changes mode in MS Word.

At the end of this letter, you will find an explanation of the changes made to the manuscript in accordance with your comments.

Once again, we wish to express our appreciation for the clear improvement of the article made possible by the reviewer and editor’s contributions. We hope the new changes meet their expectations, and we hope that they consider the work apt for publication in International Journal of Environmental Research and Public Health.

Please do not hesitate to suggest any further changes. We are at your disposal for anything else you may require.

Best regards,

Reviewer 1:

This prospective study describes 120 women in different stages of pregnancy and postpartum (first and third trimester, after delivery and 4 months postpartum) to assess the relationship between maternal neuroticism, infant’s emotional state during feeding, PTSD and worries during pregnancy.

Thank you for the opportunity to revise this interesting manuscript. There are several fundamental issues in the manuscript that I think you should address.

First – throughout the manuscript you sometimes refer to feeding and sometimes refer to breastfeeding. I assume this is a mistake and you mean feeding (due to half your sample that is not breastfeeding). However, this is very confusing to me as a reader and should be consistent throughout the manuscript.

Response: Thank you for your comment. Indeed it is a mistake. We mean “feeding”. The entire manuscript has been reviewed to correct this mistake.

A similar problem is “posttraumatic symptomatology”. You should explain what this term means. Sometimes you refer to it as PTSD and sometimes as posttraumatic symptomatology. This again is confusing to the reader. Please stay with one terminology throughout the manuscript.

Response: Thank you for your comment. Indeed, as the reviewer points out, the use of inconsistent terminology throughout the manuscript caused interpretation problems. Throughout the manuscript, we intended to refer to “posttraumatic stress disorder (PTSD)”, specifically to postpartum PTSD (P-PTSD). The entire manuscript has been reviewed, standardizing the term (“Postpartum post-traumatic stress disorder”, “Postpartum PTSD”, “P-PTSD”). The (inappropriate) term “posttraumatic symptomatology” has been removed.

Second – and this is the most important comment: you state in the introduction 2 aims for your study. The first, the association between neuroticism and infant bonding problems in feeding. The second is that worries during pregnancy have a moderating effect on the association between neuroticism and the presence of PTSD (posttraumatic symptomatology) after delivery. However, in your manuscript, you mainly focus on the second aim. Even in the conclusion, there is mainly a focus on the second aim of the study. This problem should be addressed throughout the manuscript. In order to help readers understand the flow of your manuscript subheadings may help and I think the figures should be better planned to facilitate readers understanding the models you suggest (results 3.2, 3.3 and 3.4 should be explained in better figures).

 Response: Thanks for your comments. Both objectives have been explained more clearly. Actually, objective 2 (mediation-moderate) includes objective 1 (mediation). We want to give the importance that our first objective deserves in order to be able to address the model of moderated mediation (second objective). To this end, we have reformulated our conclusions and made the appropriate modifications in the Results section.

INTRODUCTION

In lines 48-49, this sentence that associates bonding with feeding should be elaborated a little bit. The second sentence in this paragraph, line 51, why did you choose ‘therefore’?

In line 65, please explain what is posttraumatic symptomatology.

Response: Following your suggestions and reviewing the manuscript in detail, we understand that this idea that you point out is not fully developed. Therefore, we have developed in greater depth the relationship between bonding and feeding, including additional references of interest. "Therefore" term was meaningless in that sentence. The wording has been changed.

In line 65, please explain what is posttraumatic symptomatology.

Response: As indicated in the response to the reviewer's comment above,posttraumatic symptomatology” has been deleted. In this revised version of the manuscript reference is made to postpartum PTSD (P-PTSD) which is the correct term.

MATERIAL AND METHODS

285 women participated in the first part of the study (and completed the questionnaire in the first trimester). Only 120 remained in the final sample. Please give a comparison between the original cohort and the final sample. This is a possible source for major selection bias in your study.

Response: Thanks. Your suggestion is very appropriate due to the significant difference in the number of participants in both phases. We have compared the sociodemographic and clinical/obstetric characteristics of both samples. No significant statistical differences have been found. This information has been added.

In lines 126-127 – what did you mean secondary and primary studies? This is not clear.

Response: We agree with your suggestion. Thanks. Said paragraph has been rewritten using levels according to an internationally recognized system for educational levels. Specifically, “The International Standard Classification of Education (ISCED, 2011)” has been used.

Figure 1 – 3 women were excluded (did not complete the …[word missing])

Response: We regret the error in the figure. We have enlarged the box so that the text can be displayed.

Line 177 – obstetric history – how was this obtained? Through the electronic medical records?

Response: Effectively, obstetric history was obtained through the electronic medical records. This data has been incorporated.

Figure 3 – I think it is not clear and does not add a lot to the manuscript.

Response: Thanks. Indeed, Figure 3 does not provide additional information to Table 3 (now Table 4). That is why, following the reviewer's suggestion, said Figure has been eliminated.

DISCUSSION

Line 408 – groundbreaking – I suggest using a more subtle word.

Response: We have replaced that term with a more appropriate synonym (interesting).

Line 467 – I suggest deleting the first sentence (‘our study is not without limitations’). I suggest adding the selection bias (120 women out of 285 in the original cohort).

Response: We have removed the introductory sentence from the limitations paragraph. We have also added the reviewer's suggestion as a methodological limitation of the study. In general, the limitations section has been modified by adding new aspects to take into account.

CONCLUSION

Where is a statement regarding the bonding problem during feeding (aim 1)?

Response: We have included a sentence to mention the results obtained based on the first objective of the study.

Reviewer 2 Report

Well written article and well thought out study and methods.

It could use some English language edits in certain places

Intro

Line 72-73Are there any other references besides [16] to support this statement

Materials and methods

Line 123-135 could be summarized in a table if allowed or preferred

Discussion

I think this article does lack some mention of specific socioeconomic factors such as length of maternity leave, financial resources.  These factors can also influence the capacity to bond

I think the authors should mention the role that biological factors have on bonding and well being of mother and child such has hormonal fluctuations which are very dependent on the individual, serotonin fluctuations post partum.

I would also mention a limitation that this population is relatively homogenous

Author Response

Dear Evelin Li,

We would like to thank you for your interest in our manuscript entitled Mother-child bond through feeding. A prospective study including neuroticism, pregnancy worries and post-traumatic symptomatology (ijerph-2098508) in the special issue of Maternal Perinatal Mental Health 2nd Edition of the Women's Health section. We appreciate the time that you and the other reviewers have dedicated to reading the manuscript and providing suggestions. Your suggestions have enriched the manuscript considerably. Likewise, we have incorporated all the comments suggested. Following your directions, we have proceeded to revise our manuscript, highlighting the changes by using the track changes mode in MS Word.

At the end of this letter, you will find an explanation of the changes made to the manuscript in accordance with your comments.

Once again, we wish to express our appreciation for the clear improvement of the article made possible by the reviewer and editor’s contributions. We hope the new changes meet their expectations, and we hope that they consider the work apt for publication in International Journal of Environmental Research and Public Health.

Please do not hesitate to suggest any further changes. We are at your disposal for anything else you may require.

Best regards,

Reviewer 2:

Well written article and well thought out study and methods.

Response: Thank you for the positive evaluation of our manuscript.

It could use some English language edits in certain places

Response: The English language has been reviewed and modified in certain places. Thanks.

Intro

Line 72-73Are there any other references besides [16] to support this statement.

Response: New references have been added on this topic.

  1. Oddo-Sommerfeld, S.; Hain, S.; Louwen, F.; Schermelleh-Engel, K. Longitudinal effects of dysfunctional perfectionism and avoidant personality style on postpartum mental disorders: Pathways through antepartum depression and anxiety. J Affect Disord. 2016, 191, 280–288.
  2. Davies, S.M.; Silverio, S.A.; Christiansen, P.; Fallon, V. Maternal-infant bonding and perceptions of infant temperament: The mediating role of maternal mental health. J Affect Disord. 2021, 282,1323-1329.
  3. Deans, C. L. Maternal sensitivity, its relationship with child outcomes, and interventions that address it: A systematic literature review. Earl Child Develop Care. 2020, 190(2), 252–275.
  4. Vedova, A. M. D. Maternal psychological state and infant’s temperament at three months. J Reprod Infant Psychol, 2014 32(5), 520–534.

Materials and methods

Line 123-135 could be summarized in a table if allowed or preferred

Response: Following your suggestion, we have included this information in a table including the percentage of participants excluded in each phase of the study.

Discussion

I think this article does lack some mention of specific socioeconomic factors such as length of maternity leave, financial resources.  These factors can also influence the capacity to bond

Response: Thanks. We consider your comment very interesting. Since we have not considered the effect of socioeconomic factors (such as length of maternity leave or financial resources) in our model, we have included this limitation in the corresponding section.

 I think the authors should mention the role that biological factors have on bonding and well being of mother and child such has hormonal fluctuations which are very dependent on the individual, serotonin fluctuations post partum.

Response: We find the reviewer's emphasis on the biological factors associated with the development of mother-infant attachment behaviors interesting. Therefore, we put forward this suggestion as a possible future line of research, also taking into account the psychological variables that we include in our model to explain the development of the mother-baby bond. References in this regard have been incorporated.

I would also mention a limitation that this population is relatively homogenous

Response: We included your suggestion as a limitation. Thanks. 
